# Target and Task specific Source-Free Domain Adaptive Image Segmentation

**Vibashan VS**[*][1]                                                    VVISHNU2@JHU.EDU

[1] *Johns Hopkins University*

**Jeya Maria Jose Valanarasu**[*][2]                                     JMJOSE@STANFORD.EDU

[2] *Stanford University*

**Vishal M. Patel**[1]                                                  VPATEL36@JHU.EDU

**Editors:** Accepted for publication at MIDL 2024

## Abstract

Solving the domain shift problem during inference is essential in medical imaging as most deep-learning based solutions suffer from it. In practice, domain shifts are tackled by performing Unsupervised Domain Adaptation (UDA), where a model is adapted to an unlabeled target domain by leveraging the labelled source domain. In medical scenarios, the data comes with huge privacy concerns making it difficult to apply standard UDA techniques. Hence, a closer clinical setting is Source-Free UDA (SFUDA), where we have access to source trained model but not the source data during adaptation. Methods trying to solve SFUDA typically address the domain shift using pseudo-label based self-training techniques. However due to domain shift, these pseudo-labels are usually of high entropy and denoising them still does not make them perfect labels to supervise the model. Therefore, adapting the source model with noisy pseudo labels reduces its segmentation capability while addressing the domain shift. To this end, we propose a two-stage approach for source-free domain adaptive image segmentation: 1) Target-specific adaptation followed by 2) Task-specific adaptation. In the Stage-I, we focus on learning target-specific representation and generating high-quality pseudo labels by leveraging a proposed ensemble entropy minimization loss and selective voting strategy. In Stage II, we focus on improving segmentation performance by utilizing teacher-student self-training and augmentation-guided consistency loss, leveraging the pseudo labels obtained from Stage I. We evaluate our proposed method on both 2D fundus datasets and 3D MRI volumes across 7 different domain shifts where we achieve better performance than recent UDA and SF-UDA methods for medical image segmentation. Code is available at https://github.com/Vibashan/tt-sfuda

**Keywords:** Unsupervised Domain Adaptation (UDA), Source-Free UDA.

## 1. Introduction

Deep-learning based methods such as convolutional neural networks (Ronneberger et al., 2015; Zhou et al., 2018; Çiçek et al., 2016; Milletari et al., 2016; Islam et al., 2018; Valanarasu et al., 2020) and transformers (Chen et al., 2021b; Valanarasu et al., 2021; Hatamizadeh et al., 2022) have been the leading solutions for medical image segmentation. One major concern with deep neural networks (DNNs) is that they depend very much on the dataset they are trained on. DNNs trained on a particular dataset do not perform well when tested

---

[*] Contributed equally

on a different dataset even if it is of the same modality. Small changes in camera type, calibration properties, age and demographics of patients causes a distribution shift which results in a considerable drop in performance. It is very common for a domain shift to exist during deployment of medical imaging solutions in real-time (Vashist, 2017). Hence adapting a source model to a target distribution is an important problem to solve for medical imaging applications like segmentation.

Recently, many works have explored unsupervised domain adaptation (UDA) for medical image segmentation (Javanmardi and Tasdizen, 2018; Panfilov et al., 2019; Zhang et al., 2019; Yang et al., 2019; Perone et al., 2019). However, UDA methods work under the assumption that labeled source domain data and unlabeled target domain data are available during adaptation. This assumption often does not hold, particularly in medical applications, where data sensitivity and privacy concerns are crucial. Consequently, utilizing source data during adaptation, given its availability concerns, becomes problematic. Source-Free Unsupervised Domain Adaptation (SFUDA) is a established solution where we assume we have access only to the source-trained model and target data for adaptation. This setting offers a practical approach to clinical adaptation without relying on source data. Bateson et al. (Bateson et al., 2020) proposed an SFUDA method that reduces label-free entropy loss for target-domain data. Similarly, Chen et al. (Chen et al., 2021a) introduced a denoised pseudo-label technique for self-adaptation on the target data. Although these techniques facilitate model adaptation to the target domain, they often compromise the model's task-specific capabilities, such as segmentation performance, due to noisy pseudo-labels.

In this work, we propose a novel method focusing on adapting the source model to the **T**arget distribution while also making sure the **T**ask-oriented representation is preserved for **S**ource-**F**ree **U**nsupervised **D**omain **A**daptation (**TT-SFUDA**). Specifically, we introduce a two-stage pseudo-label based adaptation framework: Stage I - Target-specific adaptation and Stage II - Task-specific adaptation. In target-specific adaptation, we aim to learn target-specific representation to adapt to the target domain while generating high-quality PL. In task-specific adaptation, we aim to improve task-related (i.e segmentation) performance on target domain by leverage high-quality PL generated from Stage-I. To achieve this in Stage-I, we propose an ensemble entropy minimization loss and selective voting strategy to learn target-specific representation and generate high-quality pseudo-labels, respectively. In Stage -II, we employ a strong-weak augmentation based teacher-student self-training framework and agumentation-based consistency loss to improve the segmentation performance target domain by leveraging high-quality PL generated from Stage-I. We validate our method for many domain shifts for both 2D and 3D medical datasets and compare with recent SFUDA and UDA adaptation methods.

In summary, this work makes the following contributions: 1) We propose a new SFUDA method focusing on both target and task specific adaptation for image segmentation. 2) We introduce an ensemble entropy minimization loss and a selective voting strategy to learn target-specific representation and generate high-quality pseudo labels. 3) We propose a teacher-student self-training framework with augmentation-based consistency to improve segmentation performance on the target domain. 4) We conduct extensive experiments on both 2D and 3D data of different modalities. We show results on 7 different data-shifts for both 2D fundus images and 3D MRI volumes and outperforms recent methods.

## 2. Method

Our proposed TT-SFUDA method for medical image segmentation consists of two parts: Stage I - Target specific adaptation and Stage II - Task specific adaptation. First, we explain the base segmentation network architecture and formulate certain preliminaries and notations used in the proposed adaptation method.

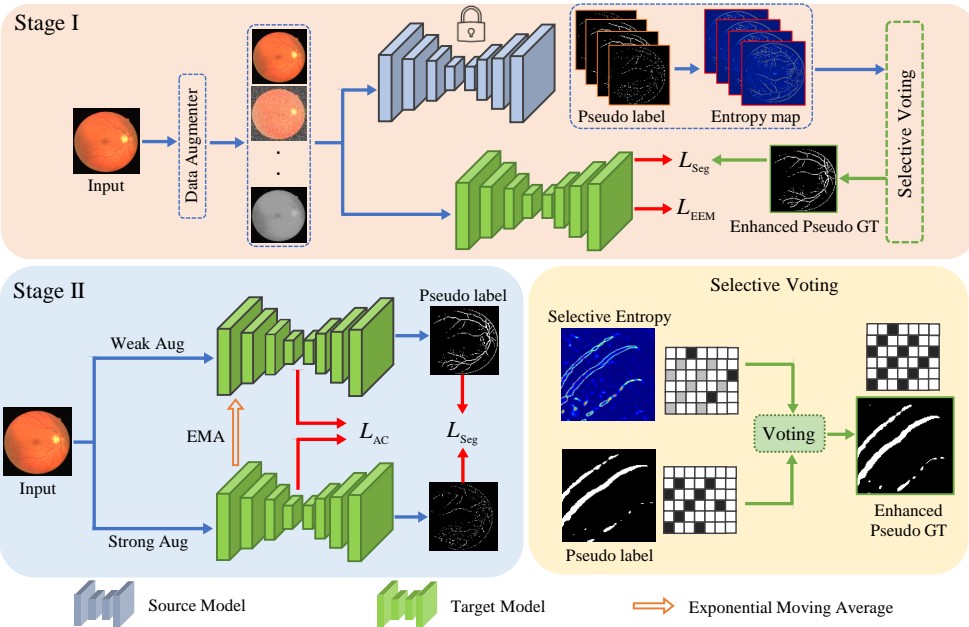

Figure 1: Overview of the proposed TT-SFUDA method. Target-specific adaptation(Stage-I) focuses on learning target-specific representation to adapt to the new domain while generating high-quality PL. Task-specific adaptation(Stage-II) focuses on improving task-related(segmentation) performance on target domain by utilizing high-quality PL generated from Stage-I

### 2.1. Preliminaries

**Base Network Details:** We use a regular UNet (Ronneberger et al., 2015) as the base segmentation network. We use a 5 level encoder-decoder framework. Each conv block in the encoder consists of a conv layer, ReLU activation and a max-pooling layer. Each conv block in the decoder consists of a conv layer, ReLU activation and an upsampling layer. For upsampling, we use bilinear interpolation. For our experiments on 3D volumes, we use a 3D UNet architecture (Çiçek et al., 2016) with same setup replacing 2D conv layer with 3D conv layers, 2D max-pooling with 3D max-pooling and bilinear upsampling with trilinear upsampling. The segmentation loss used to train this network $\mathcal{L}_{seg}$ is as follows:

$$\mathcal{L}_{seg} = 0.5BCE(\hat{y}, y) + Dice(\hat{y}, y) \tag{1}$$

**Notations:** We denote the input data as $x$ and labels as $y$. We denote the source domain distribution as $D_s = \{(x_s^n, y_s^n)\}_{n=1}^{N_s}$ and target domain distribution as $D_t = \{(x_t^n, y_t^n)\}_{n=1}^{N_t}$

where $s$ represents source, $r$ represents target, $N_s$ and $N_t$ denotes number of data-instances in source and target data respectively. We denote the source model of UNet as $\Theta_s$. The goal is to adapt $\Theta_s$ from $D_s$ to $D_t$ while using only the target data $x_t$. We assume we do not have access to $y_t$ which makes it unsupervised and also assume we do not have access to $D_s$ which makes it source-free.

## 2.2. Target-specific Adaptation

First, we address the domain shift which is the main problem that leads to the drop in performance. To perform this, we use the adaptation framework below:

**Adaptation Framework:** We define two instances of the model: the source model ($\Theta_s$) and the target model ($\Theta_t^{\text{stage1}}$). The source model, $\Theta_s$, serves as a pseudo model for generating pseudo labels. A straightforward approach to learning target-specific representations is to train the $\Theta_t^{\text{stage1}}$ model using pseudo-labels $\hat{y}_t^n$ generated from $\Theta_s$. However, this method is highly dependent on the quality of the pseudo-labels. Due to the domain shift, these pseudo-labels are typically noisy, and training the model with noisy pseudo-labels can hinder the acquisition of task-specific information, usually resulting in decreased performance. Thus, in target-specific adaptation, our initial focus is on reducing the entropy of the model to produce reliable pseudo-labels, which are then utilized to enhance task-specific representation learning. Therefore, the source model $\Theta_s$ is initialized with weights from a model trained on the source domain and is kept frozen, while the target model ($\Theta_t^{\text{stage1}}$) gradually adapts to the target domain by minimizing the Stage-I loss.

**Ensemble Entropy Minimization:** For entropy minimization, we can generate an entropy map $H(\hat{y}_t^n)$ using output prediction map $\hat{y}_t^n$ by:

$$H(\hat{y}_t^n) = -\sum p(\hat{y}_t^n) log(p(\hat{y}_t^n)) \tag{2}$$

Minimizing $H(\hat{y}_t^n)$ will result in overall minimization of model entropy. However, using just one input data to generate the entropy map might not give us all the accurate locations of high entropy. Therefore, to improve the overall entropy suppressing property of model $\Theta_t^{stage1}$, we propose an ensemble entropy minimization approach. For this, we first generate a set of augmented data $\tilde{X}_t^n = \{aug_i(x_t^n)\}_{i=1}^M$ where $M$ is the number of augmentations used. We then generate a set of pseudo labels $\tilde{Y}_t^n$ corresponding to these augmented inputs $\tilde{X}_t^n$ using the source model ($\Theta_s$). Note that $\tilde{X}_t^n$ and $\tilde{Y}_t^n$ denotes a set of data instances $\{x_t^n\}_{i=1}^M$ and $\{y_t^n\}_{i=1}^M$ depending on the number of augmentations used. Based on the generated pseudo labels $\tilde{Y}_t^n$, we generate entropy maps $H(\tilde{Y}_t^n)$ as follows:

$$H(\tilde{Y}_t^n) = -\sum p(\tilde{Y}_t^n) \log(p(\tilde{Y}_t^n)). \tag{3}$$

Now, we define the ensemble entropy minimization loss $\mathcal{L}_{EEM}$ as follows:

$$\mathcal{L}_{EEM} = \frac{1}{HW} \sum_{i=1}^{HW} (H(\hat{y}_t^n) + \frac{1}{M} \sum_{j=1}^{M} H(\tilde{Y}_t^n)). \tag{4}$$

This loss not only tries to suppress the entropy of the original pseudo-label but also of the pseudo-labels of the augmented ones making it more generalized.

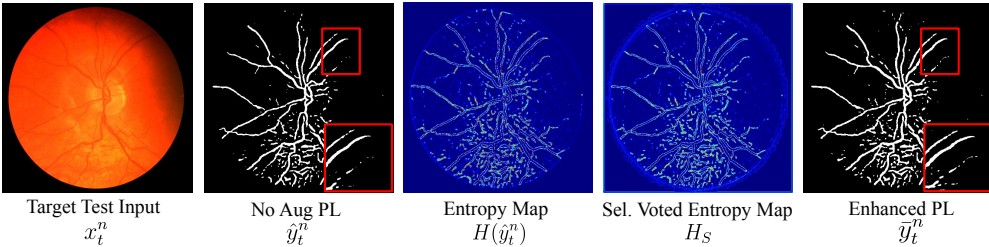

Figure 2: Steps involved in obtaining the enhanced pseudo label $\bar{y}_t^n$.

**Enhancing Pseudo-Label by Selective Voting:** Minimizing entropy is an effective strategy for adapting the model to the target domain. Furthermore, improving the quality of pseudo-labels can lead to better adaptation, as it provides direct supervision to $\Theta_t^{stage1}$. To enhance the accuracy of these pseudo-labels, which are often noisy, we propose reducing the false-negative areas within the pseudo-labels, which are typically associated with regions of high entropy. This is achieved by using the entropy maps $H(\hat{y}_t^n)$ and pseudo-labels $\tilde{Y}_t^n$, generated from augmented inputs as discussed in the preceding section. Initially, we utilize $\tilde{Y}_t^n$ to identify entropy maps that exceed a specified threshold $\delta$. This step ensures the preservation of high-entropy regions across different augmentations, highlighting predictions near the class boundary. Subsequently, we create a normalized entropy map by integrating the entropy map derived from non-augmented inputs $H(\hat{y}_t^n)$ with the selected entropy maps from augmented inputs $H(\tilde{Y}_t^n)$.

$$H_S = Norm(\alpha * H(\hat{y}_t^n) + (1 - \alpha) * H(\tilde{Y}_t^n)). \tag{5}$$

From $H_S$, we generate the selective entropy mask $(Z_t^n)$ by thresholding $H_S$ with 0.5 and we then generate a false negative mask $(U_t^n)$ as follows:

$$U_t^n = \begin{cases} 0, & p(\hat{y}_t^n) < \lambda_1 \; \& \; p(\hat{y}_t^n) > \lambda_2 \\ 1, & \lambda_1 < p(\hat{y}_t^n) < \lambda_2. \end{cases} \tag{6}$$

In practice, we find that the optimal values for $\lambda_1$ and $\lambda_2$ is between 0.25 and 0.5. Finally, we suppress the false negatives in $U_t^n$ by performing $AND$ operation with the selective entropy mask $(Z_t^n)$ and obtain enhanced pseudo-label $\bar{y}_t^n$.

We visualize the intermediate steps for a sample input $x_t^n$ in Fig. 4. It can be seen that the selectively voted entropy map $H_S$ captures more consistent high entropy regions compared to the normal entropy map $H(\tilde{y}_t^n)$. Our enhanced pseudo-label $\bar{y}_t^n$ can also be observed to be a refined version of the original pseudo-label $\hat{y}_t^n$. To train $\Theta_t^{stage1}$, we define the stage-I training loss $\mathcal{L}_{stage1}$ as:

$$\mathcal{L}_{stage1} = \mathcal{L}_{seg}(x_t^n, \bar{y}_t^n) + \mathcal{L}_{EEM}(\hat{y}_t^n). \tag{7}$$

### 2.3. Task-specific Adaptation

Although the model weights have been adapted for the new target data distribution, they do lose some task-specific information as the pseudo-labels are not perfect. Thus, target-specific adaptation to the new domain causes some unwanted hindrance to the network's

segmentation capability due to supervising on imperfect pseudo-labels. To solve this, we further refine the model in this stage to improve its task-specific learning capabilities.

**Teacher-Student Self-Training:** We use a Teacher-Student model (Wang and Yoon, 2021) where the student model $\Theta_t^{student}$ is optimized using pseudo-labels that are generated from the teacher model $\Theta_t^{teacher}$. Note that both the teacher and student are initialized with the model obtained from Stage I i.e $\Theta_t^{stage1}$. This makes sure that the pseudo-labels generated are already of less uncertainty as $\Theta_t^{stage1}$ has been trained previously to minimize entropy. The teacher model is updated by gradually transferring the weights of the continually learned student model. This is done by using exponential moving average (EMA) to update the teacher model. Thus, the teacher model can be considered as an ensemble of different student models from various training iterations.

**Augmented-guided Consistency:** To make sure that the segmentation capabilities of the model is improved, we use an augmentation-guided consistency loss while doing the teacher-student self-training. First, we perform some strong augmentations on target input data $x_t^n$ to obtain $\tilde{x}_t^n$. We also perform some weak augmentations on $x_t^n$ to obtain $\hat{x}_t^n$. $\tilde{x}_t^n$ is then passed through the teacher model to obtain $\tilde{y}_t^n$. $\hat{x}_t^n$ is passed through the student model to obtain $\hat{y}_t^n$. As the pseudo-labels generated for both these augmentations need to be consistent, we define the augmentation consistency loss $\mathcal{L}_{AC}$ as follows:

$$\mathcal{L}_{AC} = \|\Theta_{latent}^{teacher}(\tilde{x}_t^n) - \Theta_{latent}^{student}(\hat{x}_t^n)\|_2. \tag{8}$$

Note that $\Theta_{latent}^{teacher}$ corresponds to the features extracted from the latent space of $\Theta^{teacher}$ model. We extract features from the last layer of the encoder while calculating the loss $\mathcal{L}_{AC}$. The total loss for stage II $\mathcal{L}_{stage2}$ along with the pseudo-label supervision is defined as follows:

$$\mathcal{L}_{stage2} = \mathcal{L}_{seg}(x_t^n, y_t^n) + \mathcal{L}_{AC}(\tilde{x}_t^n, \hat{x}_t^n). \tag{9}$$

The final adapted model weights of stage II is represented as $\Theta_t^{stage2}$ is basically $\Theta_t^{teacher}$ after the self-training process.

## 3. Experiments and Results

**Datasets:** For 2D experiments, we focus on the task of retinal vessel segmentation from fundus images. We make use of the following datasets: CHASE (Fraz et al., 2012), RITE (Hu et al., 2013) and HRF (Odstrčilík et al., 2009).

For 3D experiments, we focus on brain tumor segmentation from MRI volumes. We make use of the BraTS 2019 dataset (Menze et al., 2014; Bakas et al., 2017, 2018) which consists of four modalities of MRI- FLAIR, T1, T1ce and T2. We study the domain shift problems between these four modalities for volumetric segmentation of brain tumor. This is a multi-class segmentation problem with 4 labels. We randomly split the dataset into 266 for training and 69 for validation.

**Implementation Details:** For 2D source experiments, we train UNet using $\mathcal{L}_{seg}(x_s, y_s)$ with an Adam optimizer and a learning rate of 0.001, momentum of 0.9. We also use a cosine annealing learning rate scheduler with a minimum learning rate upto 0.0001. For 3D experiments, we use a similar loss but is for multi-class segmentation. In TT-SFUDA experiments, for both stages, we use the Adam optimizer with a learning rate of 0.0001 and

Table 1: Results for 2D Domain shifts. Numbers correspond to dice score. **Red** and **Blue** corresponds to the first and second best performing methods respectively.

| Type | Method | CHASE→HRF | CHASE→RITE | HRF→CHASE | HRF→RITE |
|---|---|---|---|---|---|
| Source-Training | Direct Testing | 52.70 | 15.45 | 57.92 | 41.03 |
| 4*UDA | PL (Zou et al., 2018) | 53.04 ± 1.32 | 34.62 ± 1.50 | 57.41 ± 2.01 | 41.98 ± 1.67 |
|  | BEAL (Zou et al., 2018) | **60.54 ± 2.21** | 50.24 ± 2.86 | 59.21 ± 3.10 | 53.43 ± 2.51 |
|  | Output-Space (Tsai et al., 2018) | 54.21 ± 1.98 | 49.02 ± 1.82 | 57.68 ± 2.06 | 50.01 ± 2.27 |
|  | AdvEnt (Vu et al., 2019) | 58.23 ± 2.30 | 49.91 ± 2.16 | 58.50 ± 1.89 | 52.49 ± 2.11 |
| 3*SFUDA | SRDA (Bateson et al., 2020) | 55.76 ± 1.58 | 50.43 ± 1.40 | 60.81 ± 1.31 | **56.92 ± 1.99** |
|  | DPL (Chen et al., 2021a) | 56.32 ± 1.23 | **51.27 ± 1.62** | **61.29 ± 1.16** | 55.89 ± 1.23 |
|  | **TT-SFUDA (Ours)** | **58.25 ± 1.06** | **52.63 ± 1.36** | **64.95 ± 0.89** | **58.37 ± 1.03** |
| Target-Training | Oracle | 67.97 | 73.70 | 66.92 | 73.70 |

momentum of 0.9. For Stage I, we use color jitter, grayscale, color contrast etc. as different augmentation and we set hyperparameters $\alpha$, $\lambda_1$ and $\lambda_2$ to 0.75, 0.3 and 0.5 respectively. For stage II, the teacher EMA rate is set to 0.99 for each iteration in the student-teacher network. We train Stage I for one epoch and Stage II for ten epochs (Refer to supplementary for more details). The framework is implemented with Pytorch using NVIDIA TitanXp GPU.

**Performance Comparison:** We compare our proposed method TT-SFUDA against recent methods proposed for both UDA and SFUDA. Note that UDA methods have an advantage as they have access to the source train data to help the adaptation process. For UDA, we compare against a normal pseudo-label self-training (Zou et al., 2018) based approach; BEAL (Zou et al., 2018) which uses adversarial learning between source and target data with boundary information specifically for fundus images; Output-Space (Tsai et al., 2018) which involves output space projection consistency for domain adaptive segmentation and AdvEnt (Vu et al., 2019) which ensures entropy consistency between the source and target domains. For SF-UDA, we compare against SRDA (Bateson et al., 2020) and DPL (Chen et al., 2021a). SRDA uses a task prior while DPL uses an uncertainty-guided denoised pseudo-label based approach. Note that we adopt these methods for 3D by making necessary changes. We leave out BEAL and Outer-Space for 3D comparison as they have been specifically designed for 2D. More details about this can be found in supplementary material. In Tables 1 and 2, we tabulate our results for domain shifts in 2D fundus images and 3D MRI volumes respectively. We achieve better performance compared to both UDA and SFUDA baselines across almost all shifts. We present sample qualitative results in Fig. 3 where we observe a good performance for TT-SFUDA when compared to other methods.

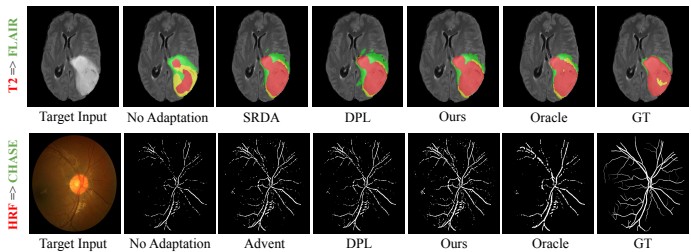

Figure 3: Qualitative results for samples on fundus and BraTS dataset.

Table 2: Results for 3D Domain shifts. Numbers correspond to dice score reported in the following order: Whole Tumor/Tumor Core/Enhancing Tumor. **Red** and **Blue** corresponds to the first and second best performing methods respectively.

| Type | Method | T2 –>FLAIR | T1 –>T1ce | FLAIR –>T2 |
|---|---|---|---|---|
| Source-Training | Direct Testing | 52.60/13.70/37.25 | 30.96/12.33/21.08 | 43.11/12.79/34.84 |
| 3*SFUDA | ENT (Vu et al., 2019) | 54.51/14.07/38.50 | 32.40/13.38/21.99 | 44.01/12.83/35.27 |
| | PL (Zou et al., 2018) | 52.95/13.80/37.38 | 31.04/12.46/21.17 | 43.23/12.67/34.79 |
| | SRDA (Bateson et al., 2020) | 57.23/14.02/38.91 | **33.98**/13.76/**22.67** | 43.17/12.52/**35.27** |
| | DPL (Chen et al., 2021a) | **58.32/14.10/41.13** | 33.62/**14.08/22.35** | **44.93/13.04/35.86** |
| | **TT-SFUDA (Ours)** | **59.06/14.16/40.67** | **34.21/13.95**/22.18 | **44.46/12.83**/34.83 |
| Target-Training | Oracle | 75.50/26.57/57.27 | 53.28/13.63/35.26 | 83.95/22.40/51.51 |

Table 3: Ablation analysis for Stage-wise experiments for 2D Domain shifts.

| Type | Method | CHASE→HRF | CHASE→RITE | HRF→CHASE | HRF→RITE |
|---|---|---|---|---|---|
| Source-Training | Direct Testing | 52.70 | 15.45 | 57.92 | 41.03 |
| 3*SFUDA | Stage I | $56.17 \pm 1.08$ | $27.19 \pm 1.35$ | $60.18 \pm 1.10$ | $48.07 \pm 1.32$ |
| | Stage II | $54.62 \pm 1.52$ | $39.28 \pm 2.13$ | $58.40 \pm 1.17$ | $52.97 \pm 1.63$ |
| | Stage II → Stage I | $55.20 \pm 1.45$ | $18.28 \pm 2.26$ | $63.67 \pm 1.70$ | $49.02 \pm 2.08$ |
| | Stage I → Stage II | $\mathbf{58.25 \pm 1.06}$ | $\mathbf{52.63 \pm 1.36}$ | $\mathbf{64.95 \pm 0.89}$ | $\mathbf{58.37 \pm 1.03}$ |
| Target-Training | Oracle | 67.97 | 73.70 | 66.92 | 73.70 |

**Ablation Study:** Table 3 presents the results of different combinations of stage-wise experiments on 2D domain shifts. The first two rows correspond to individual stage adaptation performances. The last two rows correspond to different sequences of stage-wise adaptation performances. From the first two rows, we can observe that Stage II is performs better than Stage I when there is a huge gap between direct testing and oracle performance. Performing task-specific adaptation makes the model focus more on segmentation rather than learning target-specific representation which essentially leads to noise overfitting. Therefore, when there is a large domain shift (CHASE→HRF), Stage II has more room for task-specific improvements but fails to learn target-specific representation in case of a small domain shift (CHASE→RITE). This is clearly observed in Stage II → Stage I experiment, where first learning task-specific representation essentially overfits the model to the noise generated from the pseudo-labels. Consequently, when we perform entropy minimization for target-specific adaptation, the model performance drops by a large margin. Moreover, we can observe that the drop in Stage II → Stage I is directly proportional to the difference in the domain gap. Therefore, target-specific adaptation followed by task-specific adaptation produces best performance as seen from the Stage I → Stage II experiment.

## 4. Conclusion

In this work, we proposed a new method for source-free unsupervised domain adaptive image segmentation called TT-SFUDA. We perform TT-SFUDA in two stages: Stage I- Target specific adaptation and Stage II - Task specific adaptation. In the first stage, we learn target-specific representation and generate high-quality pseudo-labels. In the second stage, improve the segmentation performance by leveraging a self-training framework with augmentation-guided consistency loss. Evaluation on multiple 2D and 3D domain shifts datasets show the effectiveness of our method compared to recent UDA and SFUDA methods.

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

## Appendix A. Discussion:

The target and task-specific adaptation for source-free unsupervised domain adaptation can be extended to any SF-UDA medical setting. Moreover, the adaptation performance can be effectively increased by setting appropriate hyperparameters. In detail, the target-specific can be performed for multiple epochs if the target dataset has a huge domain shift. However, it should be performed for one or two epochs if the dataset size is small or the target dataset contains a small domain shift. Therefore, setting optimal learning parameters for different datasets will ensure optimal target-specific adaptation.

Similarly, for task-specific adaptation setting appropriate hyperparameters for the student-teacher framework (Gou et al., 2021) can essentially learn optimal task-specific representation. Further, in task-specific adaptation, the model can be trained to learn task-specific representation continuously. However, by performing early stopping (Prechelt, 1998), one can fully exploit the task-specific adaptation. To this end, we train Stage I for one epoch and Stage II for ten epochs.

For both Stage I and Stage II, all loss weights were set to 1. In selective voting strategy, the false negatives lie around the PL threshold(0.5). Thus, $\lambda_1$ and $\lambda_2$ are 0.3 and 0.5 and experimentally, we observed within 0.3-0.5, the probability of false-negative is high. For strong augmentation, we perform ColorJitter, RandomGrayscale, RandomSolarize, and RandomAutocontrast.

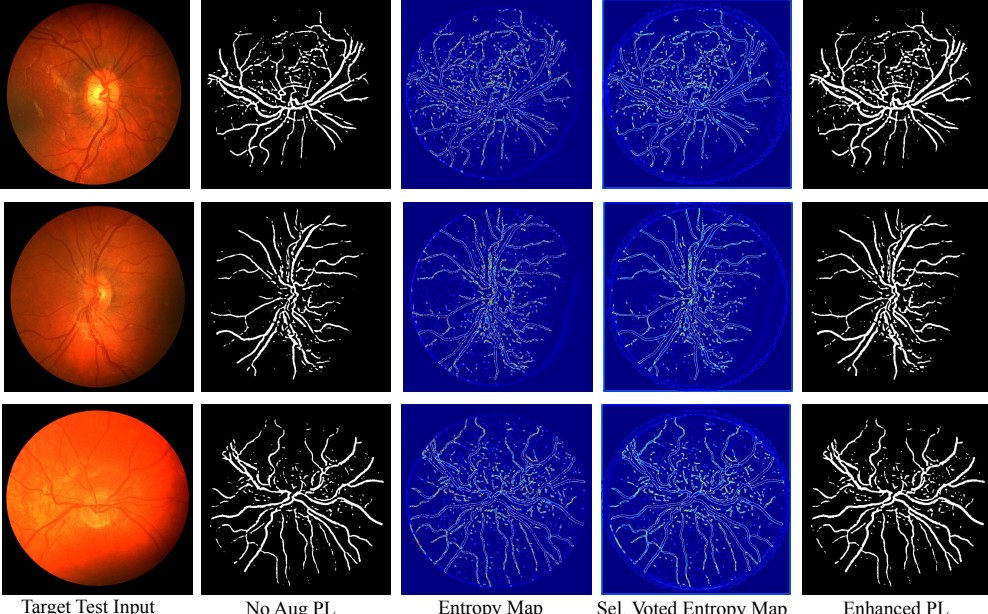

| Target Test Input | No Aug PL | Entropy Map | Sel. Voted Entropy Map | Enhanced PL |

Figure 4: More qualitative visualization of proposed approach to enhance pseudo-label generation $\bar{y}_t^n$.

# Appendix B. More details on the datasets

To give a better clarity about the datasets we use, we provide more details about them in Tables 4 and 5.

| 2D Datasets | Ages | Resolution | #Data |
|:---:|:---:|:---:|:---:|
| CHASE | 5 to 16 | 999x960 | 28 |
| RITE | 25 to 90 | 768x584 | 40 |
| HRF | 21 to 90 | 3504x2336 | 18 |

Table 4: Meta-details of the 2D datasets.

| 3D Datasets | Modality | #Data |
|:---:|:---:|:---:|
| BraTS-T1 | T1 | 335 |
| BraTS-T2 | T2 | 335 |
| BraTS-FLAIR | FLAIR | 335 |
| BraTS-T1ce | T1ce | 335 |

Table 5: Meta-details of the 2D datasets.

