# OpenReview forum: "Target and task specific source-free domain adaptive image segmentation"
_MIDL.io/2024/Conference — MIDL 2024 Poster_

### Official Review · Reviewer_rTyX · 2024-02-27

**Confidence:** 4
**Preliminary Rating:** 3
**Recommendation:** Poster
**Final Rating:** 4

**Summary:**

In this paper, the authors present a novel method for unsupervised domain adaptation in medical image segmentation task. The research question is of great importance for the medical image analysis community considering large amounts of unlabeled data. The experimental results are convincing in showing the presented model at least being strongly competitive with SOTA frameworks under the designed scenarios.

**Strengths:**

1. UDA is an important research direction for medical image analysis. This paper presents strong potential value to the community.
2. The paper clearly articulates the framework designs and experiments, providing convincing evidence for model effectiveness.
3. The two-stage framework and selective voting strategy are novel in design, especially the selective voting idea that are potentially applicable to future research with similar pipelines.

**Weaknesses:**

1. In stage I, the inclusion of original pseudo label entropy in loss is confusing.
2. The description to the selective voting idea is clear, but lacking experimental results to back up its effectiveness.
3. The same goes with the $L_{ac}$ loss in the second stage. I understand this design as aiming to generalize latent space representation under diverse augmentations. But penalizing latent space may also cause some numerical problems in training. Please consider adding evidence to support the design choice.
4. The framework relies on another segmentation model with reasonable performance to start. If initial segmentation fail significantly, the framework may fail to detect and get trained on misleading pseudo labels.

**Detailed Comments:**

1. In equation (4), the subscript $j$ to $H$ is misleading for me, please consider removing it.
2. The paper applies two types of augmentation in experiments, strong and weak. The description for strong augmentation is clear, please consider adding details for the weak augmentation.

**Justification Of Final Rating:**

I would like to thank the authors for providing comprehensive rebuttals answering all my concerns. Based on the clarifications and additional information, I am more than willing to improve my rating for this paper.

**Justification Of The Preliminary Rating:**

The paper clearly articulates the method and shows convincing evidence. But the loss function applied in stage I appears to contradict with the designed pipeline. It seems a major flaw to me. Happy to change my rating if this part is explained.

**Questions To Address In The Rebuttal:**

1. In section 2.2, the authors argue that the loss function jointly suppress entropy in original and augment pseudo labels. Considering the model weights of original model is fixed during training, how is the loss going to suppress entropy in labels? What is the point of including a fixed-weight model output into the loss function.

2. Please consider adding evidence for the effectiveness for selective voting and $L_{ac}$. Also, how difficult it is to tune the threshold parameter $\lambda_1, \lambda_2$ in your experiments?

**Special Issue:**

No

---

> ### Author Response · Authors · 2024-03-18
> **Official Reply to Reviewer rTyX**
>
> ***1) Stage I inclusion of pseudo-label entropy:***
>
> In the SFUDA, we adapt a source-trained model without access to the original source data and due to domain shifts, initial pseudo-label predictions often exhibit high uncertainty. In response, Stage I of our method prioritizes the generation of reliable pseudo-labels for the subsequent Stage II, which involves student-teacher training. To enhance the reliability of these pseudo-labels, we specifically aim to reduce their inherent uncertainty. This is achieved by applying various data augmentations, which serve to expand the target data, and by systematically minimizing the entropy across all corresponding predictions. Overall, the need for original pseudo-label entropy is designed to minimize the uncertainty associated with our pseudo-labels from the source-trained model, enabling a reliable pseudo-label generation process.
>
> ***2) Subscript is misleading:***
>
> Thank you for your suggestion. We will update it in our final version to avoid confusion.
>
> ***3) Strong and weak augmentation:***
>
> For strong augmentation, we perform ColorJitter, RandomGrayscale, RandomSolarize, and RandomAutocontrast. For weak augmentation, we just perform Randomflip. We will update this detail in our final version.
>
> ***4) Section 2.2 Frozen weight and loss update:***
>
> Thank you for pointing out the confusion in Section 2.2  and we apologize for the misunderstanding. As illustrated in Figure 1, we have two models: the source-trained model (frozen) and the Stage-I model (trainable). Since we don't have access to the source data, we always maintain a frozen source-trained model to ensure the preservation of the source domain information. The Stage-I model is initialized with weights from the source-trained model and gradually adapts to the target domain by minimizing the Stage-I loss. Therefore, the loss is backpropagated only to the Stage-I model, while the source-trained model remains frozen throughout the training process. We will clarify this detail and update Section 2.2 in the revised version.
>
> ***5) Penalizing latent space evidence and motivation:***
>
> The loss in the latent space is motivated by self-supervised methods like SimCLR[1], BYOL[2], and MoCo[3]. These approaches demonstrate that models learn robust features when they produce similar representations for different augmentations of the same image by minimizing the loss at latent space. Applying this principle, our consistency loss aims to enhance the segmentation ability of both teacher and student models by ensuring they extract comparable and robust features from varied augmented inputs. This method is grounded in proven self-supervised learning techniques that improve model generalization and robustness.
>
> ***7) Parameter senstive:***
>
> For both Stage I and Stage II, all loss weights were set to 1. Furthermore, our experiments on the variation in the $\lambda_1 - \lambda_2$ range had a negligible impact on performance, specifically within $\pm$ 1%. Overall, this indicates that while selective voting can effectively reduce false negatives, fine-tuning of the threshold parameter, especially within this critical range, is nuanced but not overly sensitive. We will clarify these detail in our revised version.
>
> ***8) Initial segmentation fail:***
>
> Thank you for your insightful comment. I agree that our framework, like other Source-Free Unsupervised Domain Adaptation (SF-UDA) methods, depends on a reliable initial segmentation model. If the initial model is weak or the domain gap is large, our approach could struggle, similar to other SF-UDA methods. This issue is a common challenge in SF-UDA methods due to their dependency on pre-trained models and assumptions regarding domain similarity [4]. We acknowledge this limitation and believe that exploring ways to enhance model robustness against large domain discrepancies under reasonable settings is a promising direction.
>
>
> ***Reference:***
>
> 1. Chen, Ting, Simon Kornblith, Mohammad Norouzi, and Geoffrey Hinton. "A simple framework for contrastive learning of visual representations." In International conference on machine learning, pp. 1597-1607. PMLR, 2020.
>
> 2. Grill, Jean-Bastien, Florian Strub, Florent Altché, Corentin Tallec, Pierre Richemond, Elena Buchatskaya, Carl Doersch et al. "Bootstrap your own latent-a new approach to self-supervised learning." Advances in neural information processing systems 33 (2020): 21271-21284.
>
> 3. He, Kaiming, Haoqi Fan, Yuxin Wu, Saining Xie, and Ross Girshick. "Momentum contrast for unsupervised visual representation learning." In Proceedings of the IEEE/CVF conference on computer vision and pattern recognition, pp. 9729-9738. 2020.
>
> 4. Mehra, Akshay, Bhavya Kailkhura, Pin-Yu Chen, and Jihun Hamm. "Understanding the limits of unsupervised domain adaptation via data poisoning." Advances in Neural Information Processing Systems 34 (2021): 17347-17359.

---

### Official Review · Reviewer_5jne · 2024-02-28

**Confidence:** 2
**Preliminary Rating:** 4

**Summary:**

This paper proposes a two-stage approach: 1) Target-specific adaptation and 2) Task-specific adaptation. Stage-I improves pseudo-label quality with an entropy minimization loss and selective voting. Stage-II enhances segmentation using teacher-student training and consistency loss. Our method outperforms recent UDA and SF-UDA for medical image segmentation.

**Strengths:**

-  Introducing a two-stage approach for SFUDA provides a structured framework for addressing domain shift in medical image segmentation.
- Outperforms UDA and SF-UDA methods on 2D fundus datasets and 3D MRI volumes across 7 different domain shifts highlights its effectiveness and generalizability.

**Weaknesses:**

- The study did not demonstrate a significant enhancement in results for 2D domain shifts.
- To strengthen the findings, it is recommended to incorporate statistical significance into the results, providing a more robust assessment of the outcomes.

**Detailed Comments:**

- Performing bootstrapping on the test results would enhance the reliability and generalizability of the findings.

**Justification Of The Preliminary Rating:**

The study did not show a substantial improvement in results for 2D domain shifts, suggesting the need for further validation. Incorporating statistical significance into the results would offer a more robust assessment of the outcomes, strengthening the findings. Additionally, performing bootstrapping on the test results could enhance their reliability and generalizability. To expand the scope and applicability of the proposed method, it would be beneficial to test it on chest X-rays (2D) or CT images, in addition to the fundus images (2D) that were already evaluated. This broader testing could provide insights into the method's performance across different types of medical imaging data.

**Questions To Address In The Rebuttal:**

Could be nice to try the proposed method for testing with chest X-rays (2D) or CT images, in addition to the tested fundus images (2D).

---

> ### Author Response · Authors · 2024-03-18
> **Official Reply to Reviewer 5jne**
>
> ***1) More statistical significance for results:***
>
> We appreciate the reviewer's suggestion. Regarding statistical significance, the standard deviations associated with our method across all domain shifts are consistently lower or comparable to those of the best-performing methods. This suggests that TT-SFUDA is not only effective but also stable. Furthermore, it is noteworthy that, unlike typical UDA methods which assume access to both source and target data, our method assumes access only to source-trained data, making it challenging. Nonetheless, the standard deviation is lower than that of most UDA methods, such as BEAL and Advent. Specifically, our TT-SFUDA method significantly outperforms others, with notable improvements in the dice score from 56.32 ± 1.23 (DPL, the second-best in SFUDA) to 58.25 ± 1.06 (TT-SFUDA) in the CHASE→HRF shift, and from 56.92 ± 1.99 to 58.37 ± 1.03 in the HRF→RITE shift, indicating superior performance with less variability. We include these details in the final version of our paper.
>
> ***2) 2D chest X-rays or CT images:***
>
> Thank you for your suggestion. We acknowledge the potential value of testing with chest X-rays or CT images. However, the fundus datasets such as HRF, CHASE_DB1, and RITE present unique challenges due to their extremely small size, comprising 18, 40, and 28 images, respectively. This makes them highly challenging test sets for source-free domain adaptation compared to CT and X-ray datasets. While our current focus is on demonstrating our method's effectiveness under these highly challenging conditions; Including X-rays and CT images represents a promising direction for future work to gain more insights.
>
> ***3) Did not demonstrate a significant enhancement in 2D results:***
>
> We respectfully disagree with the comments regarding the lack of significant enhancements in results for 2D domain shifts. Our TT-SFUDA model demonstrates substantial improvements over Direct Testing, with increases in Dice scores of +5.5 (CHASE→HRF), +37.1 (CHASE→RITE), +7.0 (HRF→CHASE), and +17.3 (HRF→RITE). Additionally, when compared to traditional UDA methods, our approach exhibits superior performance. Notably, it outperforms the leading UDA method, BEAL, in three out of the four domain shifts, despite not utilizing source data during adaptation.

---

### Official Review · Reviewer_JEQu · 2024-03-03

**Confidence:** 3
**Preliminary Rating:** 4
**Recommendation:** Poster
**Final Rating:** 4

**Summary:**

The authors proposed a new SFUDA method for domain adaptation without accessing source data, which is useful in medical imaging field due to data privacy. The study focused on not only domain adaption to the target dataset but also image segmentation-specific adaptation by introducing entropy minimization and selective voting for better pseudo-labels. They also incorporated the teacher-student model for better tuning to help increase model performance.

**Strengths:**

1. The authors have tested their method in various datasets, including 2D and 3D images with different modalities. This increases the validity and reliability of the work.
2. The authors have conducted comprehensive ablation studies to prove the use of different stages proposed by the authors.

**Weaknesses:**

1. Overall, the paper is well-written, but there are certain areas where a bit more clarification would enhance understanding.
2. There is no validation set for the hyper-parameter tuning. This will increase reliability as the authors could choose the best-performing model in the validation set to compare across different methods.
3. In Table 2, the proposed method outperforms most of the other methods, but in FLAIR to T2 domain shift, it does not perform the best. The authors could discuss more and give insights on why this is the case.

**Detailed Comments:**

1. The paper already contains this information, but for better clarity, the authors could provide separate tables for the description of the datasets that are used for evaluation. For example, a brief description of the dataset, modality, and number of total images. This could also lead to a better interpretation of the performance which could be attributable to the size difference of the dataset. For example, if the dataset is huge, it might already contain some images with similar data distribution to the smaller dataset making it easier for the domain adaptation.

**Justification Of Final Rating:**

The authors have addressed the questions raised in previous comments: (1) provided the descriptive statistics of the dataset (2) explained why there is no validation dataset (3) explained why the domain adaptation is not optimal in all cases. Overall, the authors provided a novel approach for domain adaptation in target specification and task specification and evaluated the approach in multiple datasets of different modalities.

**Justification Of The Preliminary Rating:**

The authors did well in the experiment design, testing on various datasets and modalities, and proved that the method outperformed the baseline methods in most of the domain shift scenarios. The authors also conducted ablation studies to prove the necessity of every part of the method framework. However, the authors could provide more explanation and discussion on the performance results, and also some more context of each dataset that is used for testing.

**Questions To Address In The Rebuttal:**

1. The authors performed a comprehensive ablation study and delivered a complete discussion on the difference in the performance. However, there are several domain shifts scenario that the proposed method did not perform better than the other methods (Table 2) and the authors did not further explain why this happens or the factor they believe that might play a role in this. For example, is it because of the inherent propagation of variance of the proposed method framework since there are many "moving parts" in the pipeline? Or is it because of the challenge of the specific type of domain shift that the authors are doing (e.g. FLAIR to T2 domain adaptation is generally hard and all methods have already reached the limit so that the difference in the performance do not have statistical significance)?

**Special Issue:**

No

---

> ### Author Response · Authors · 2024-03-18
> **Official Reply to Reviewer JEQu**
>
> ***1) More details on the datasets:***
>
> We thank the reviewer for the comment and for better clarity, we have now included tables that gives details about the modality, demographic details, as well as the number of data in the appendix. The tables can also be seen here:
>
> | 2D Datasets |   Ages   | Resolution | #Data |
> |:-----------:|:--------:|:----------:|:-----:|
> |    CHASE    |  5 to 16 |   999x960  |   28  |
> |     RITE    | 25 to 90 |   768x584  |   40  |
> |     HRF     | 21 to 90 |  3504x2336 |   18  |
>
> | 3D Datasets | Modality | #Data |
> |:-----------:|:--------:|:-----:|
> |   BraTS-T1  |    T1    |  335  |
> |  BraTS-T2   |    T2    |  335  |
> | BraTS-FLAIR |   FLAIR  |  335  |
> |  BraTS-T1ce |   T1ce   |  335  |
>
> ***2) Reason for not using validation set:***
>
> We understand that having a validation set for hyper paramter tuning would increase the reliability. However, we would like to point out that the datasets that we work on are very small in number and splitting them further into train/val/test would lead to very less number of data for training as well as validation. For example, HRF dataset only has 18 images and doing a 70/15/15 split would result in only having 2 images for validation which might make the experiments very dependent on those 2 images chosen for validation.
>
> ***3) Reasons for why our method might not be optimial for certain shifts:***
>
> While our method peforms better overall for most domain shifts, we do agree that for some cases our performance is not the best. However, we would like to point out that our performance is still comparable to the best performance for FLAIR->T2 domain shift. We believe this observation is because of the nature of certain domain shifts being difficlut for our proposed method to address. We also found that certain shifts like FLAIR -> T1 are very difficult and if the source training performance is itself very low, it becomes very difficult for any domain adaptation method to achieve a notable gain in performance.

---

### Meta-Review · Area_Chair_tD26 · 2024-04-04

**Recommendation:** Accept (Poster)
**Confidence:** 4

**Metareview:**

This paper proposes a source-free unsupervised domain adaptation method for segmentation using a 2 stage approach: target-specific adaptation followed by task-specific adaptation.

Strengths:
+ UDA is important research direction, and source-free is particularly relevant for medical imaging domain
+ Novel two-stage approach for handling the problem
+ Good experiments with multiple datasets and ablation studies

Weaknesses:
- No validation data used for hyper parameter tuning
- While standard deviations were given, significance testing was not performed, and there are differing opinions regarding whether the results show significant increase in performance compared to other SOTA

Following rebuttal, the reviewers felt that the strengths generally outweighed the weaknesses. While there were some concerns regarding experimental results, overall I think this paper would be of high interest to the MIDL community, and therefore recommend accept.

---

### Decision · Program_Chairs · 2024-04-06

Accept (Poster)